# Modeling Wildland Firefighter Travel Rates by Terrain Slope: Results from GPS-Tracking of Type 1 Crew Movement

**Patrick R. Sullivan [1],\*, Michael J. Campbell [1]  , Philip E. Dennison [1]  , Simon C. Brewer [1]   and Bret W. Butler [2]**

[1]   Department of Geography, University of Utah, Salt Lake City, UT 84112, USA; mickey.campbell@geog.utah.edu (M.J.C.); dennison@geog.utah.edu (P.E.D.); simon.brewer@geog.utah.edu (S.C.B.)

[2]   Rocky Mountain Research Station, USDA Forest Service, Missoula, MT 59808, USA; bret.butler@usda.gov

\*   Correspondence: patrick.sullivan@utah.edu

**Abstract:** Escape routes keep firefighters safe by providing efficient evacuation pathways from the fire line to safety zones. Effectively utilizing escape routes requires a precise understanding of how much time it will take firefighters to traverse them. To improve this understanding, we collected GPS-tracked travel rate data from US Interagency Hotshot "Type 1" Crews during training in 2019. Firefighters were tracked while hiking, carrying standard loads (e.g., packs, tools, etc.) along trails with a precisely-measured terrain slope derived from airborne lidar. The effects of the slope on the instantaneous travel rate were assessed by three models generated using non-linear quantile regression, representing low (bottom third), moderate (middle third), and high (upper third) rates of travel, which were validated using *k*-fold cross-validation. The models peak at about a −3° (downhill) slope, similar to previous slope-dependent travel rate functions. The moderate firefighter travel rate model mostly predicts faster movement than previous slope-dependent travel rate functions, suggesting that firefighters generally move faster than non-firefighting personnel while hiking. Steepness was also found to have a smaller effect on firefighter travel rates than previously predicted. The travel rate functions produced by this study provide guidelines for firefighter escape route travel rates and allow for more accurate and flexible wildland firefighting safety planning.

**Keywords:** firefighter safety; escape routes; hiking travel rates; topography

## 1. Introduction

Suppressing and managing wildland fires places firefighters, the most fundamental and important resource for fire management [1], into potentially life-threatening situations. Between the "great fires" of 1910 [2,3] and 2017, 1128 firefighter fatalities occurred while on the job [4]. Of these fatalities, approximately 44% occurred because of entrapment, which the National Wildfire Coordinating Group (NWCG) defines as "a situation where personnel are unexpectedly caught in a fire behavior-related, life-threatening position where planned escape routes or safety zones are absent, inadequate, or compromised" or burnover, which NWCG defines as "an event in which fire moves through a location or overtakes personnel where there is no opportunity to utilize escape routes and safety zones" [5].

A common theme in both entrapments and burnovers is the inability to utilize escape routes and safety zones. Escape routes are pre-defined paths from a firefighter's location to a planned area of refuge from fire danger, termed a safety zone. These two tools are a part of the Lookouts, Communications, Escape routes, and Safety zones protocol (LCES) [6], a series of four critical safety

measures employed by fire crews throughout the US designed to keep firefighters safe and prevent injuries and fatalities. Each component of LCES needs to be established before firefighting begins and reevaluated as conditions change [7]. Implementation of LCES is done by firefighters in the field, predominantly based on ground-level information. The tactical decisions made in establishing and reevaluating LCES can be hindered by human factors such as insufficient knowledge of surroundings and conditions, inexperience, overextension of resources, and/or loss of situational awareness [8–11]. The risk of being entrapped or burned-over is increased when firefighters are faced with these barriers [12,13]. Page et al. [14] found that entrapment incidents are likely substantially underreported, which would indicate that these barriers may be more of a problem than currently acknowledged. In an effort to reduce entrapment and burnover events, recent research has been aimed at better understanding the identification, evaluation, and designation of safety zones and escape routes.

When escape routes are defined, the path of least resistance is used as a guiding factor and firefighter safety is the end goal. Firefighters must have a thorough understanding of both their own physical ability to travel across the local terrain and fire behavior in order to maintain a positive margin of safety–that is, to be able to reach a safety zone before the fire can reach the crew [15]. While fire behavior has been extensively researched and modeled e.g. [16–18], comparably little effort has been directed at quantitative definitions of attributes that define effective escape routes. More research is needed to understand the efficiency with which firefighters are able to traverse escape routes under various landscape conditions, including the slope of the underlying terrain.

Re-creations and personal accounts of individual fire events that led to injuries or fatalities made up much of the firefighter safety literature until recently. MacLean [19] detailed the events and circumstances that resulted in the death of 13 smokejumpers during the Mann Gulch fire in 1949. Rothermel [20] analyzed the fire behavior, wind conditions, fuel, and gave a minute-by-minute update of the crew locations and travel rates for the Mann Gulch fire. Putnam [10] analyzed the South Canyon fire that led to 14 fatalities, documenting the firefighters' travel rates, the terrain conditions, rate of fire spread, and weight of personal equipment. Through reenactments, Putnam [10] found that the firefighters traveled anywhere between 0.3 and 1.7 m/s depending on terrain, with extremely fast reenactment traveling at 2.1 m/s. Butler et al. [21] used the data from these two events to produce a relationship between the firefighter travel rate and terrain slope. This study suggested that firefighters moving across flat (0%) and up low (10–20%), moderate (20–40%), and steep (40–60%) slopes travel at rates of roughly 1.3, 0.9, 0.6, and 0.3 m/s, respectively. While these historical re-creations are useful learning tools and represent foundational knowledge of firefighter movement, they lack the quantitative precision afforded by controlled studies of firefighter travel rates. Two controlled experiments looking at how firefighters travel along escape routes were carried out in the early 2000s. Ruby et al. [22] looked at the physiological effects of traveling with and without a pack along a trail, and Alexander et al. [23] tested the effects of slope, load weight, vegetation type, vegetation density, and trail type on firefighter travel rate. Both studies found load to have a negative effect on travel rates; however, neither produced predictive models.

If the effects of landscape conditions on firefighter travel rates are known, and those landscape conditions can be mapped in advance of a fire event, then fire crews can be directed along routes with known transit times. To that end, Campbell et al. [24] developed the Escape Route Index, which relies on slope-travel rate relationships to map relative egress capacity on the landscape scale. If both the fire's rate of spread and a fire crew's rate of travel can be accurately predicted, then threats to firefighter safety can be minimized through geospatial modeling [25]. Fryer et al. [26] developed a theoretical model for predicting spatial evacuation triggers using fire spread and travel rates, but relied on a pedestrian travel rate function that is not calibrated to firefighters.

Recent work has focused on the use of remote sensing and geographic information science (GIS) to improve the identification and evaluation of safety zones [27,28]. There have been several mathematical functions developed for predicting pedestrian travel rates, particularly as they are affected by terrain slope [29–35]. These functions were not derived from data specific to firefighters, and assume no load

carriage. Firefighters are likely to be carrying loads exceeding 15 kg (33 lbs), and studies have found that carrying a load has a significant effect on travel rate [22,23,36,37]. Previous pedestrian travel rate models also fail to capture the unique physical characteristics of firefighters, whose fitness and experience traversing diverse terrain are likely greater than nonfirefighters. Additionally, current travel rate functions model human movement under normal (e.g., recreational) circumstances. These models fail to capture the speed at which firefighters can traverse an escape route in an emergency situation. Better travel rate functions over complex terrain and integration of this information into a safety zone assessment have been identified as wildland firefighter safety research needs [14]. For these reasons, we generated a new, flexible set of models to predict firefighter travel rates under varying slopes, improving the capacity for estimating escape route travel time and potentially reducing the risk of injury or fatality to firefighters.

## 2. Materials and Methods

### 2.1. Data Collection

A request for help with data collection was sent out to US Interagency Hotshot Crews around the country. Eleven Interagency Hotshot Crews (IHCs) consisting of 20–22 people each agreed to participate in this study. Location data were collected at a one-second interval using Qstarz BT-q100XT global positioning system (GPS) units attached to Type 1 firefighters in IHCs as they trained. Forty-five GPS units were distributed among the crews to maximize participation throughout the training season, which took place from late March to early June, 2019. This resulted in nine of the 11 participating IHCs receiving seven GPS units each, and the other two IHCs receiving 20 GPS units each. In addition to the GPS units, Hotshot crews were provided with armbands to carry the GPS units and data collection sheets. Before using the GPS units, firefighters were asked to complete a data collection sheet containing their name, crew name, squad identifier, job, GPS ID, age, gender, height, weight, and typical load weight. Once this information was recorded, firefighters took their assigned GPS into the field and recorded their movement during training hikes.

Before collecting any data, firefighters were instructed to turn their GPS unit on and wait to allow the GPS to acquire a reliable signal from satellites to improve positional accuracy. Then, the firefighters began their hike. When the activity was complete, the firefighters were asked to turn their GPS unit off. This process was repeated until GPS unit memory was full or until the next participating crew was scheduled to receive units, at which point GPS units were mailed to project personnel at the University of Utah for data downloading and processing.

### 2.2. Data Processing

Since crew members hiked along the same trails together as a group, GPS points recorded among crew members should follow a nearly identical path at approximately the same time. A review of the GPS data confirmed this spatiotemporal clustering. Therefore, GPS point positions that deviated from a trail were due to GPS positional error. If GPS positional error is spatially random, then recorded points along a trail should be distributed around the "true" trail in a Gaussian fashion, with the highest density of points being close to the trail, and a decreasing density of points radiating outward (Figure 1a). Thus, we can assume that the highest point densities values represent the "true" position of the trail. Accordingly, using a combination of high-resolution imagery and point density of the GPS data, the trails firefighters traveled upon were digitized in ArcGIS [38] (Figure 1b). The distance between each point and the high-resolution digitized trail was measured to gain an understanding of each point's positional error. All points were then snapped to the nearest corresponding point along the digitized trails (Figure 1c). Snapping all points to the trails corrected for any GPS positional error perpendicular to the trail by recording a better approximation of the true location of each point, while maintaining the original temporal data.

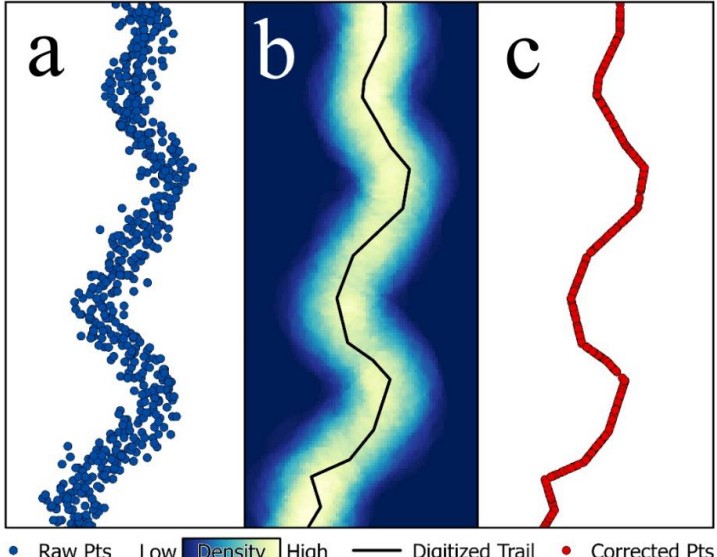

**Figure 1.** (**a**) Raw global positioning system (GPS) points representing instantaneous locations of firefighters as they traverse along a single-track trail, with positional error; (**b**) point density raster dataset used to digitize a line representing the average GPS position and the "true" trail; (**c**) corrected GPS points with locations snapped to the trail.

Once the points were snapped, the elevation of each corrected point needed to be found. In order to get the most accurate elevation data, we opted to use airborne lidar data. Lidar is a form of active remote sensing that generates highly-precise three-dimensional models of terrain and above-ground structures by emitting laser pulses and measuring the time it takes a pulse to return to the sensor [39]. Accordingly, lidar is an ideal source of high-resolution terrain slope information. Freely-accessible lidar data were not available for all of the areas where crews trained; therefore, only trails and associated GPS data with lidar data were used for analysis. Next, 25-centimeter spatial resolution digital terrain models (DTMs) were generated using lidar data acquired from the United States Geologic Survey's 3D Elevation Program and LAStools lidar data processing software [40]. DTM elevation was extracted for each corrected GPS point.

Slope and travel rates were calculated between each corrected GPS data point and the preceding point. Using the DTM-derived elevation, the slope was calculated using the equation:

$$slope = \sin^{-1}\left(\frac{\Delta d_z}{\sqrt{\Delta d_x{}^2 + \Delta d_y{}^2}}\right),\tag{1}$$

where $\Delta d_z$ is the change in lidar-derived elevation from the current to the previous point in m, and $\Delta d_x$ and $\Delta d_y$ are the change in horizontal distance from the current to the previous point in the $x$ (east-west) and $y$ (north-south) directions in m, respectively. For the most accurate reflection of true distance traveled, three-dimensional (slope-adjusted) distance ($\Delta d_{xyz}$) was calculated as follows:

$$\Delta d_{xyz} = \sqrt{\Delta d_x{}^2 + \Delta d_y{}^2 + \Delta d_z{}^2}.\tag{2}$$

Instantaneous rate of travel ($v$; in ms-1) was calculated between each GPS point using the equation:

$$v = \frac{\Delta d_{xyz}}{\Delta t},\tag{3}$$

where $\Delta t$ is the change in time from the current to the previous point, in s.

To ensure that the highest quality of data were being used as the basis of modeling, an intensive data screening process was carried out (Figure 2). All data filtering and modeling were performed in R to ensure processing efficiency and reproducibility [41]. First, all points that were initially greater than 10 m away from the digitized trail were removed. Even with the corrected (trail-snapped) point location, the positional error associated with these points reduced the likelihood that they were "snapped" to the correct location along the digitized trail, thus introducing error into the travel rate calculation. Since we were interested in modeling movement, all points with travel rates of 0 m/s were excluded. GPS units were sometimes left on, continuing to record when firefighters entered vehicles and drove away from hikes. The subsequent travel rates from these records were very high. High travel rates also resulted between points with extreme positional inaccuracy. This problem created an artificially large distance between the two points over the one-second interval, and therefore, a falsely high travel rate. Since firefighters did not record any sprints, we decided to filter out all speeds over 7 m/s, or approximately 15.7 mph, to get rid of these erroneous points, as this threshold represents the limits of reasonable pedestrian movement. Next, points with slopes less than −45° and greater than +45° were excluded due to extreme difficulty in hiking or running across these terrains, and poor representation in the dataset. Then, a three-point moving average of travel rate over time was calculated for each activity. All points with a travel rate greater than two standard deviations away from the mean were eliminated to reduce the influence of outliers. Firefighters often left their GPS units on while standing still, before or after an activity, or during a break. Although stationary, points recorded by the GPS units varied spatially due to GPS positional error. This resulted in very slow, artificial travel rates when there should be none. To address this problem, all points where the running average was less than 0.25 m/s, or approximately 0.6 mph, for more than 10 consecutive points (10 s) were excluded. GPS units take time to acquire satellite signals and record accurate locations. Activities that lasted for 100 s or less were filtered out to get rid of data where the GPS units were warming up for a large portion of the total activity.

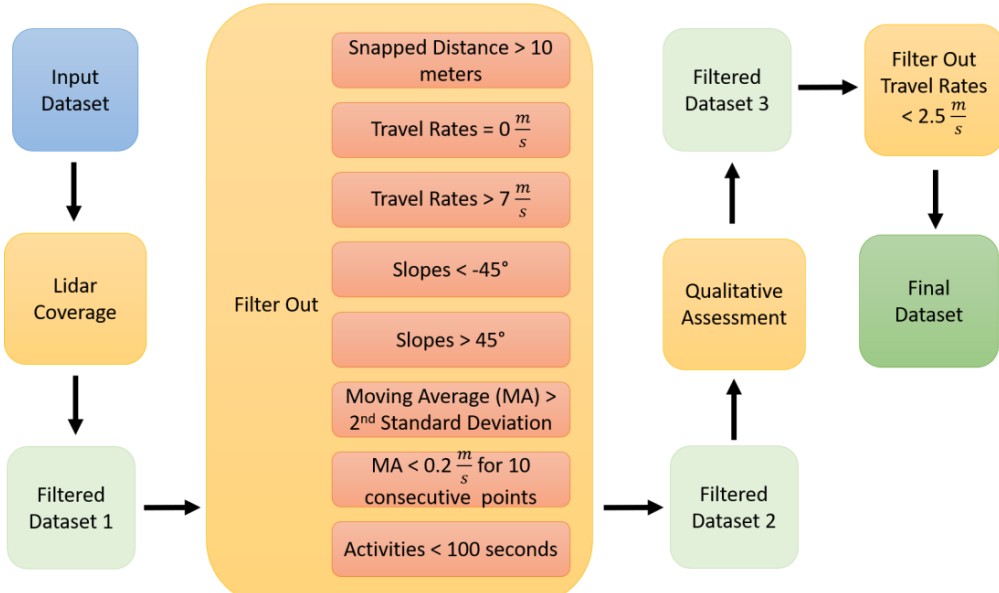

**Figure 2.** Flowchart visualizing the data filtering processes. The blue square represents the input data, orange rectangles show processes, light green squares represent intermediate datasets, and the dark green square represents the final dataset.

While crews were asked to limit their activities during GPS logging to exclusively either hikes (under load) or runs, activities of some crews appeared to contain a mix of both. Once the data filtering was completed, slope-travel rate records were qualitatively and visually assessed for their adherence to

a "typical" slope-travel rate relationship at the individual activity level. Those that broadly resembled typical relationships (i.e., faster movement on flatter slopes, slower movement on steeper slopes) were retained. Those that did not (e.g., variable movement on flatter and steeper slopes) were assumed to be the result of some combination of unique training practices that do not represent normal hiking conditions or mixing running and hiking within the same activity. The data from these atypical activities were removed from consideration.

Some points with high travel rate values remained after the qualitative filtering. According to a number of the superintendents whose crews were involved in data collection, intermittent running during a hike is not uncommon while training. The high threshold of 7 m/s meant to filter out erroneously high travel rates missed these points. Travel rates above 2.5 m/s (approximately 5.6 mph) were filtered out of the dataset to ensure all points included were recorded while hiking.

*2.3. Analysis*

The largest and most recent pedestrian travel rate study was crowd-sourced through the fitness app Strava [29]. Campbell et al. [29] took travel rate data from 29,928 individuals representing 421,247 individual activities and fit a range of percentile-based slope-travel rate functions ranging from slow hiking (1st percentile) to very fast running (99th percentile), comparing the relative predictive accuracy of three different probability density function forms (Laplace, Gauss, and Lorentz). They found that the Lorentz function form was best at estimating slope-travel rate relationships across a wide range of percentiles:

$$v_{Lorentz} = c\left(\frac{1}{\pi b\left[1 + \left(\frac{\theta - a}{b}\right)^2\right]}\right) + d + e\theta. \tag{4}$$

The interior of this equation is a general Lorentz function where *a* centers curve (the slope of peak travel rate), *b* controls the width of the curve (how sharply travel rates are affected on either side of the slope of peak travel rate), and $\theta$ is the independent variable (the slope of the terrain, in degrees). Campbell et al. [29] added three additional terms to the Lorentz function to accurately fit travel rate data. Given that probability distributions result in a value bounded by 0 and 1, but travel rate data are not bounded by 1, a multiplicative term *c* was introduced. Since all of the travel rates are greater than 0 m/s, an additive term *d* was added. Finally, the slope does not affect the human travel rate evenly across all slopes. As the uphill slope increases, travel rates consistently decrease. Conversely, traveling downhill initially increases travel rates, but eventually, travel rates start to slow down at a muted rate as the slope decreases further. To account for this anisotropy, a slope-dependent additive term *e* was added to the function.

The Lorentz function in Equation (4) was fit to the filtered firefighter data using non-linear quantile regression in R with the quantreg package [42], which utilizes the method described by Koenker and Park [43]. Quantile regression was used because of the inherently variable nature of human travel rates. One model cannot accurately represent hiking travel rates on a given slope for every firefighter under all possible conditions. Instead, we decided to model low, moderate, and high travel rate quantiles through quantile regression. In other words, the slowest tertile (0th–33rd percentile), middle tertile (34th–66th percentile), and fastest tertile (67th–100th percentile) of the dataset were modeled. This was done by modeling the middle percentile within each tertile of the data, which come out to be percentiles at 16.7 (1/6), 50 (3/6), and 83.3 (5/6). This roughly corresponds to modeling the mean percentile and the first standard deviation around the mean.

*k*-fold cross-validation was utilized to assess the predictive accuracy of each model. However, firefighters generally hiked together in groups over the same trail. If two firefighters walked together on a trail, the recorded travel rates would be roughly the same. To avoid building and validating models with these pseudo-replicate points, a *k*-means clustering algorithm was used to cluster segments of a trail based on the UTM coordinates for each GPS point. Points that were recorded on a clustered

segment of a trail were assigned to the same fold. Five clusters/folds ($k = 5$) were used in this process. Mean absolute error (MAE), $R^2$, and bias were calculated for each iteration, then averaged.

## 3. Results

Out of the 11 IHCs that agreed to collect data, 85 firefighters carried GPS units. Two of these units came back broken, two units were not returned, four units were returned without required consent paperwork for University of Utah's Institutional Review Board (IRB), and all of the units returning from one IHC came back corrupted. In total, 732 h, or almost 31 days, worth of data were collected at elevations ranging from 263 to 2418 m above sea level. Lidar availability eliminated training areas used by 5 crews, and filtering further reduced the data such that 21 firefighters, one female and 20 males (Table 1), from three crews contributed data to the final dataset, which consisted of 108,600 records. These firefighters collected data for 44 hikes over 9 different trails, totaling 209 km. The shortest recorded hike was 0.3 km, the longest hike was 12.7 km, and the average hike was 4.8 km. The average positional drift from the actual trail was 1.5 m.

**Table 1.** Demographics of firefighters that collected data used in the final analysis.

|  | Age | Weight (lbs/kg) | Height (ft/m) | Load (lbs/kg) |
|---|---|---|---|---|
| **Average** | 29 | 179.7/81.5 | 5.9/1.8 | 50/22.7 |
| **Maximum** | 43 | 215.0/97.5 | 6.3/1.9 | 85/38.6 |
| **Minimum** | 21 | 140.0/63.5 | 5.4/1.6 | 40/18.1 |

The Lorentz functions for the three modeled tertiles are shown in Figure 3, all of which produced significant coefficients (Table 2, Equation (4)). The low travel rate function peaked at 0.974 m/s at a slope −2.8°, the moderate travel rate function peaked at 1.404 m/s at −2.9°, and the high travel rate function peaked at 1.699 m/s at −2.6°.

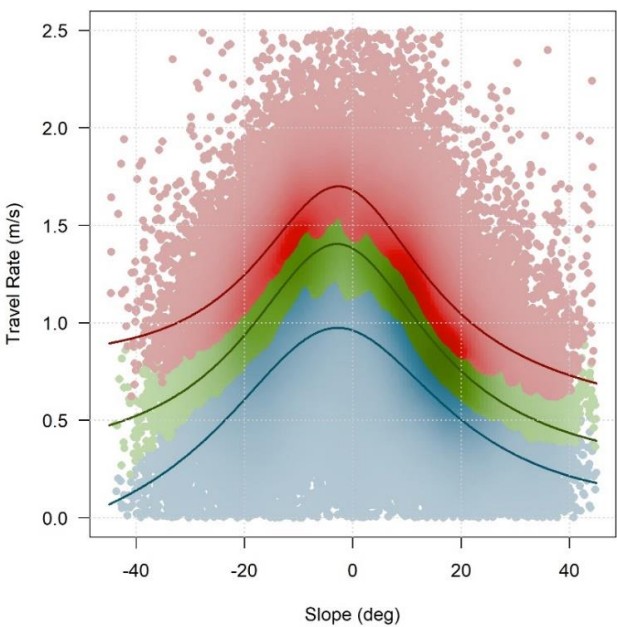

**Figure 3.** Filtered hiking travel rate data against slopes with the low, moderate, and high travel rate functions. Points are colored by model and shaded based on the density of points at a given location on the chart. Points and associated predictive function lines in the low tertile are colored blue, moderate tertile are colored green, and high tertile are colored red.

**Table 2.** Model Coefficient Summary, with * representing significance at α = 0.01.

| Modeled Tertile | a | b | c | d | e |
|:---:|:---:|:---:|:---:|:---:|:---:|
| Low | −3.3717 * | 25.8255 * | 92.6594 * | −0.1624 * | 0.0019 * |
| Moderate | −2.8292 * | 20.9482 * | 77.6346 * | 0.2228 * | −0.0004 * |
| High | −2.2893 * | 19.4024 * | 65.3577 * | 0.6226 * | −0.0020 * |

Similar to most previous pedestrian travel rate functions, the low, moderate, and high firefighter travel rate functions achieved their highest travel rates on slightly negative (downhill) slopes, as explained by the negative coefficient *a* (Table 2). Travel rate generally decreased with increasing slope for all three functions (Figure 3). The width parameter, *b*, decreased from the low to the high tertile, indicating a stronger slope effect as the travel rate increased. Values of *e*, the coefficient that accounts for anisotropy in hiking rates on either side of the peak, were close to zero but positive for the low travel rate function and negative for other two functions (Table 2). The expected negative coefficient indicates that downhill travel on steeply negative slopes was faster than uphill travel on steeply positive slopes for the moderate and high travel rate functions. The positive value of *e* for the low travel rate function may be due to poorer fit of the model and sparser data on the steepest slopes.

5-fold cross-validation was run on each model to measure the predictive accuracy of the models (Table 3, Figure 4). Among the three models, the moderate travel rate model performed the best, with a relatively low MAE (0.081 m/s) and relatively high $R^2$ (0.86). This makes sense given that points tend to be normally distributed by the slope, with the highest density of point occurring around the 50th percentile (Figure 3). Therefore, the variance within the middle tertile (the center of the distribution) was more constrained than the upper and lower tertiles (the tails of the distribution). The low travel rate model featured the lowest performance of the three (MAE = 0.172 m/s; $R^2$ = 0.39), likely attributable to the abundance of near-zero travel rates across the entire range of slopes in the lower tertile. The high travel rate model fell in between the others in terms of explanatory power and predictive error (MAE = 0.152 m/s; $R^2$ = 0.57). All three models had bias values close to 0 m/s, indicating the models are unbiased, and are not prone to consistent overestimation or underestimation of travel rates. Likelihood ratio tests were performed on each model and we found that all models were significantly different from the null hypothesis of a no slope-travel rate relationship.

**Table 3.** Model Cross-Validation Summary.

| Modeled Tertile | MAE | $R^2$ | Bias |
|:---:|:---:|:---:|:---:|
| Low | 0.172 m/s | 0.39 | 0.046 m/s |
| Moderate | 0.081 m/s | 0.86 | 0.002 m/s |
| High | 0.152 m/s | 0.57 | −0.047 m/s |

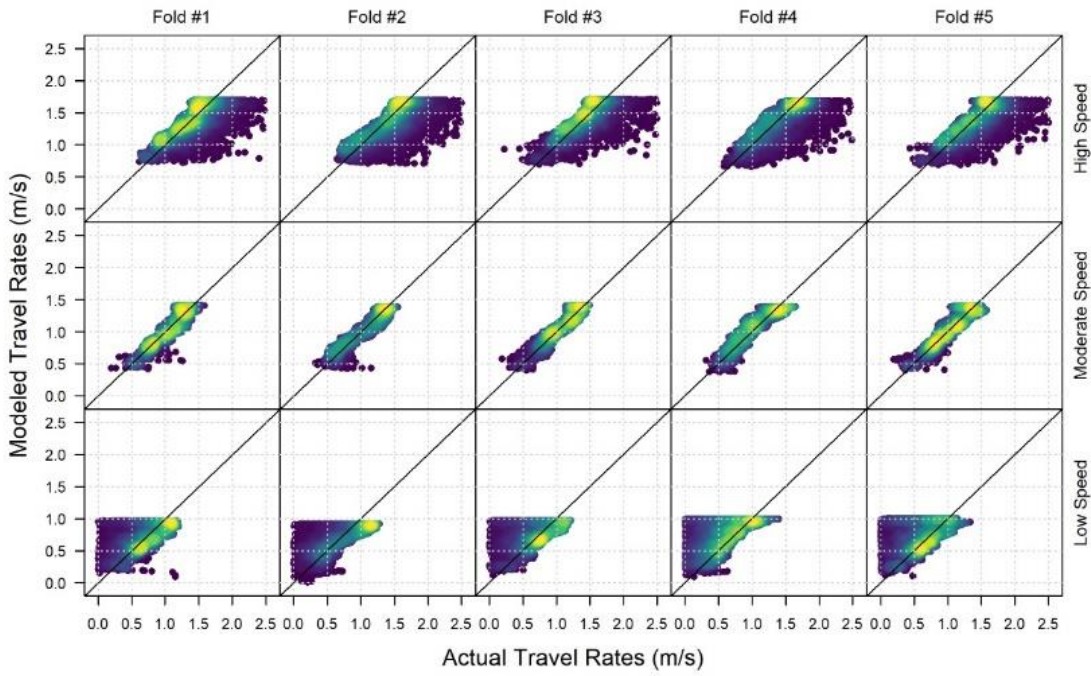

**Figure 4.** Actual versus modeled travel rates for the low, moderate, and high tertiles for each fold in k-fold cross validation. Density of points is shown on a blue-to-yellow color scale, where yellow represents the highest density.

## 4. Discussion

Out of the three models produced by this study, the moderate travel rate model performed the best with an $R^2$ value of 0.86 and a MAE of 0.08 m/s. While the other two models did not fit the broader range of data in the low and high tertiles as well, they are still useful for predicting above- and below-average travel rates and times. With MAEs of 0.172 m/s and 0.152 m/s for the low and high models, respectively, the difference in predicted travel rate and actual travel rate is not drastically different. Additionally, the lack of bias within the models indicates the remaining error is noise that is irreducible without more information. When these models are used to predict travel times, the error may not be substantial over short distances but would amplify as distances increase.

Quantile regression was chosen because of its ability to build a flexible model, accounting for the natural variability in human travel rates. While we modeled three percentiles, this approach could easily be expanded to include a higher number of percentiles to choose from. However, the increased number of models would come at the cost of a greater difficulty in choosing the best model to apply to a given situation. For example, if a crew superintendent had 20 models to choose from and a crew member was slowed by injury, should they use the 10th, 15th, or 20th percentile model to predict their crew's travel rate along an escape route? Additionally, the more models added, the more inflated the accuracy measures become, as the variance within each quantile shrinks. For these reasons, we chose to only model the three percentiles which roughly correspond to the mean percentile and the first standard deviation around the mean. We believe this is a good balance between practicality and flexibility, and not over-inflating reported model quality.

It is informative to compare these newly-derived travel rate functions to relevant, existing functions. Tobler's hiking function [31] has been applied in a variety of contexts for predicting slope-controlled travel rates, including wildland firefighter safety [26]. Campbell et al. [29]'s slope-travel rate function was used as the basis of predicting egress in the Escape Route Index [24]. Irmischer and Clarke [35]'s function is based on the movement of military cadets, who may possess similar travel rate characteristics to the firefighting population. A comparison reveals that these functions predict travel rates mostly higher than those predicted by the low firefighter travel rate function, but below those predicted

by the moderate firefighter travel rate function (Figure 5). An exception is Tobler's hiking function, which predicts much faster travel rates on low slopes and at its peak comes close to the high firefighter travel rate function.

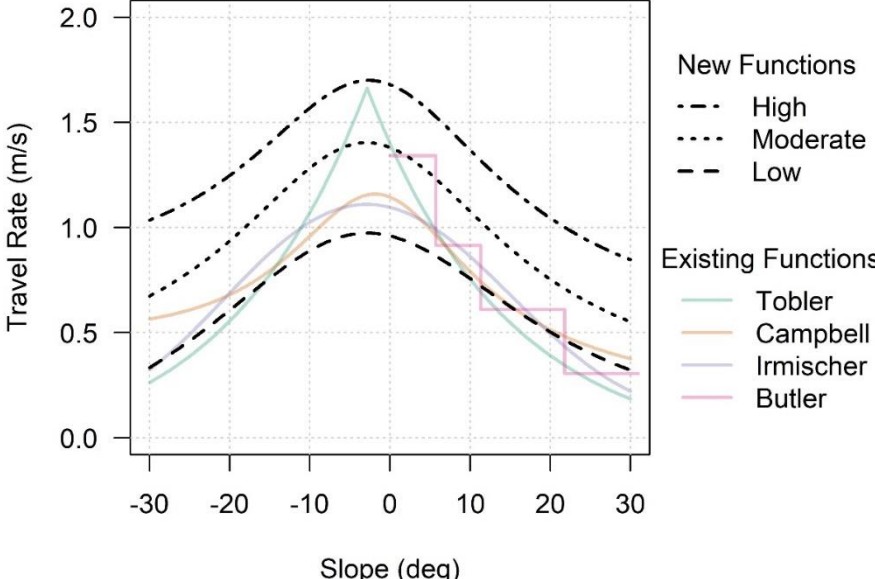

**Figure 5.** Firefighter travel rate functions (black lines) in comparison to four existing, relevant travel rate functions. The travel rate function shown form Campbell et al. [29] represents their 5th percentile function, which they demonstrated to be aligned with typical pedestrian hiking rates.

Since the moderate function predicts travel rates greater than recent pedestrian models from Campbell et al. [29] and Irmischer and Clarke [35], Type 1 firefighters may hike faster on average than the general population and military cadets. Past models may be better at predicting aggregated travel rates for an activity with breaks, but not as good at predicting instantaneous travel rates. The analysis of the Mann Gulch and South Canyon incidents by Butler et al. [21] provided guidelines for firefighter travel rates up positive slopes shown in Figure 5. For positive slopes steeper than approximately 5 degrees, these guidelines more closely agree with the low travel rate function. For flatter slopes, the Butler et al. [21] recommendation is in closer agreement with the moderate travel rate function.

All travel rate functions predict a decrease in travel rate as slope increases. However, this effect is less pronounced in the new firefighter travel rate functions compared to previously published functions (Figure 5). Firefighters could have less of a performance decrease on steeper slopes than the general public due to both the physical fitness of firefighters and their training. The average person does not regularly hike long distances or travel on steep surfaces. Therefore, the average person may be more vulnerable to the effect slope has on decreasing speed. In contrast, firefighters are in peak physical shape and train to travel through steep terrain regularly. Accordingly, if one were to use a non-firefighter-specific travel rate function for predicting escape route travel time on steep slopes, the travel time would likely be overestimated.

These new functions can be applied in a variety of contexts, including GIS-based least-cost path modeling for escape route planning based on existing terrain conditions. In the absence of such a capacity, more general guidelines can also be derived, such as those seen in Table 4 below. Based on the moderate travel rate function, a firefighting crew could expect to travel one kilometer in approximately 12 min, adding approximately three minutes for a downhill 15° slope and adding approximately six minutes for an uphill 15° slope. The more conservative low travel rate function would predict 17 min for a one kilometer hike, adding approximately five minutes for a downhill 15° slope and approximately nine minutes for an uphill 15° slope. Additional guidelines can be derived from Tables 2 and 4.

**Table 4.** Travel time estimates, in minutes, for the three wildland firefighter travel rate functions (low, moderate, and high), based on two different fixed travel distances (1 mile and 1 km), across five different fixed slopes (−30°, −15°, 0°, +15°, and +30°).

| Travel Rate | Travel Distance | Slope | | | | |
|---|---|---|---|---|---|---|
| | | *−30°* | *−15°* | *0°* | *+15°* | *+30°* |
| *High* | *1 mi* | 25.9 | 19.1 | 16.0 | 22.5 | 31.7 |
| | *1 km* | 16.1 | 11.9 | 9.9 | 14.0 | 19.7 |
| *Moderate* | *1 mi* | 39.8 | 24.1 | 19.4 | 29.8 | 48.6 |
| | *1 km* | 24.7 | 15.0 | 12.1 | 18.5 | 30.2 |
| *Low* | *1 mi* | 80.3 | 35.4 | 27.9 | 43.0 | 83.2 |
| | *1 km* | 49.9 | 22.0 | 17.4 | 26.7 | 51.7 |

Although this study modeled travel rates while participants carried heavy loads, the load effects were not explicitly modeled. This was justified based on how the study participants moved; the Type 1 firefighters that were modeled generally travel in a group. While group size may vary, the travel rate of the group is dependent on the slowest member of the group, regardless of the load they bear. However, it should be noted that travel rate predictions made from the functions presented in this study assume load carriage. In an emergency situation, where firefighters drop their packs and run to evacuate, these functions no longer apply and travel rates would be expected to increase. Additionally, if an orderly escape breaks down and crew members get separated, group travel can no longer be assumed and the functions may have a reduced value.

Firefighters in training are expected to be in the same physical condition they will maintain throughout the fire season. Consequently, the data collected throughout the study are assumed to be similar to how crews travel during the fire season and are representative of real firefighting situations. However, fire season had not yet begun for the crews collecting data. Future research should examine the effect fatigue has on travel rates, and could examine how the number of days fighting fire, temperature, time of day, trail surface conditions, length of the activity, panic, or the effort put into the activity affect travel rates.

The results of this study should be considered in the broader context of a suite of recent research into the powerful infusion of geospatial technology into wildland fire management and firefighter safety. Improving our understanding of how firefighters traverse various landscapes is essential for the assessment of relative egress capacity [24], estimation of medical evacuation time [44], mapping of least-cost escape routes [34], and knowledge of when to trigger an evacuation [26]. However, escape routes are of little use unless they direct fire crews towards suitable safety zones, which can be geospatially identified and evaluated [27,28]. The designation of both escape routes and safety zones should also take into account hazards such as the potential for falling snags, a significant cause of firefighter fatality [45]. To ensure that firefighters are not only kept safe, but are able to accomplish their fire management goals with maximal efficiency, these spatial analysis tools should be used in conjunction with spatially-explicit metrics such as the suppression difficulty index [46,47], and the identification of potential control locations [48].

## 5. Conclusions

This study modeled the effect of a slope on the travel rate while Type 1 firefighters hiked carrying a load. These models show that, on average, firefighters hike faster and more consistently across varying slopes than previous functions derived from a more general, non-firefighter population, predicted. While the primary focus of this study was to model travel rates for Type 1 firefighters, the models produced could be applied to personnel with similar experience and fitness levels such as search and rescue teams, park rangers, or even experienced hikers and backpackers. Although ground surface

condition was not explicitly measured, the travel rate data gleaned from this study were gathered from firefighters traversing a variety of gravel and dirt roads and trails. This makes the predictive functions presented in this study applicable in a diverse range of environments both within and beyond the wildland fire environment.

This study is a part of a larger effort to improve the safety conditions for wildland firefighters. To effectively utilize escape routes, the rate at which firefighters can travel across a given route must first be understood. Although effects such as fatigue, panic, and trail quality were not included in the study, with these newly developed models, wildland firefighter personnel do not need to rely on intuition or over-generalized travel rate models to accurately predict how long it would take a group of firefighters to travel along an escape route to safety. The results of this study can be used by firefighters to choose a high, moderate, or low travel rate model to predict travel times depending on the situation at hand. For instance, a superintendent may opt to use the low travel rate model if a crew member is injured, or if there is some other factor slowing the crew's rate of travel. Alternatively, the high model may be selected if a fire is encroaching on the crew and they need to get to safety faster.

**Author Contributions:** Conceptualization, P.R.S., M.J.C., P.E.D., S.C.B., and B.W.B.; Methodology, P.R.S., M.J.C., P.E.D., and S.C.B.; Software, P.R.S. and M.J.C.; Validation, P.R.S. and M.J.C.; Formal Analysis, P.R.S. and M.J.C.; Data Curation, P.R.S., M.J.C., and P.E.D.; Writing—Original Draft Preparation, P.R.S.; Writing—Review and Editing, P.R.S., M.J.C., P.E.D., S.C.B., and B.W.B.; Visualization, P.R.S. and M.J.C.; Funding Acquisition, P.E.D., M.J.C., B.W.B. All authors have read and agreed to the published version of the manuscript.

**Funding:** This research was funded by the USDA Forest Service National Fire Plan through the Washington Office of the Forest Service Deputy Chief for Research, and the National Wildfire Coordinating Group Fire Behavior Subcommittee, Cooperative Agreements 18JV11221637153 and 19JV11221637142.

**Conflicts of Interest:** The authors declare no conflict of interest.

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
