# Peer review of "Modeling Wildland Firefighter Travel Rates by Terrain Slope: Results from GPS-Tracking of Type 1 Crew Movement"

_fire, doi:10.3390/fire3030052_

Round 1

Reviewer 1 Report

The authors took up a very interesting problem. They investigated the aspect of rescue team speed at different angles of slopes. It is useful to plan the action or to input proper data into simulation software.

However, a revision is required before the publication:

  1. I put "Significance of Content" as "low" because the results are very specific - only wildland is in interest. I suggest writing some sentences about how the results can be used in different types of terrain.
  2. "We assumed that crew members were hiking along the same trail."- that's a weak point. Possibly in the same way rescue team going as a group would have a different speed than a lonely rescuer. The authors should prove that their assumption was right.
  3. There is a coma after eq.1 and dot after eq.2. The manuscript should be revised in terms of the edition.
  4. According to Table 3, the results are referred to 1km of the distance. The table can not be in the Conclusions section, and this information should be introduced in the Method chapter.
  5. Figure 3 indicates that velocity will be the same up and down under the same angle. It is not the truth.
  6. Tested people should be introduced by weight, age, height, load, and any other important parameters. Measurements should be carried out for different loads. It would give real times as a general conclusion. Without this comparison, they are very specific and this is the second weakness of the text.
  7. I'm not sure about the term "travel rate". The unit is m/s so it is travel speed or velocity. "The rate" in physics is usually referred to other quantity parameters - mass or volume flow, i.e. airflow rate is m3/s.

Reviewer 2 Report

GPS-tracked travel rate data from several US Hotshot crews was collected during training, i.e., tracked while hiking and carrying standard loads along trails with known terrain slopes. The effects of slope on travel rate were assessed by non-linear quantile regression, representing low (bottom third), moderate (middle third), and high (upper third) travel speeds. The moderate firefighter travel rate model mostly predicts faster movement than previous slope-dependent travel rate models, suggesting that firefighters generally move faster than non-firefighting personnel while hiking. Slope effects were found to have less effect on firefighter travel rates than previously predicted. The travel rate functions established provide guidelines for firefighter escape route travel rates and may allow for more improved wildland firefighting safety planning.

Title

The title is representative for the study.

Abstract

The Abstract describes what is done, is well written and it presents the main conclusions of the study. However, some more information on the speed being highest for about slopes of about -2° may be added, and that this result is in agreement with previous studies.

General comments

This is a well done piece of research. I appreciate that you supplied numbers also in metric units throughout, i.e. good for the international "metric" community.

Just one question: Why is High in bold face in Table 3 (while the other speeds are not)?

Final comment

Thank you for a very interesting read.

Round 2

Reviewer 1 Report

The authors revised the text according to the suggestions. Now, it can be published.